# Identification and Evaluation of qRT-PCR Reference Genes in *Melanaphis sacchari*

**DOI:** 10.3390/insects15070522

**Published:** 2024-07-11

**Authors:** Kunliang Zou, Tonghan Wang, Minghui Guan, Yang Liu, Jieqin Li, Yanlong Liu, Junli Du, Degong Wu

**Affiliations:** 1College of Agriculture, Anhui Science and Technology University, Chuzhou 233100, China; zkl151231@163.com (K.Z.); 18767401521@163.com (T.W.); wlhljq@163.com (J.L.); liuyl@ahstu.edu.cn (Y.L.); wudg@ahstu.edu.cn (D.W.); 2Anhui Province International Joint Research Center of Forage Bio-Breeding, Chuzhou 233100, China; mhbelucky@163.com; 3College of Resources and Environment, Anhui Science and Technology University, Chuzhou 233100, China; 18298115131@163.com

**Keywords:** *Melanaphis sacchari*, reference gene, stability, Normfinder

## Abstract

**Simple Summary:**

The sorghum aphid, a significant pest affecting sorghum cultivation in several continents, feeds primarily on sorghum leaves and stems by sucking sap. This feeding behavior results in the depletion of nutrients, sugars, and water in the plants, leading to inhibited growth, chlorosis, wilting, and ultimately death. The main objective of this study was to assess the stability of potential reference genes in the sorghum aphid (*Melanaphis sacchari*). Nine candidate reference genes underwent a rigorous selection process, and their reliability was evaluated using qRT-PCR Ct values. Various experiments were conducted to determine the best reference genes for the sorghum aphid through systematic stability analyses. The research offers a dependable collection of reference genes that can improve studies on gene expression and functional genomics related to the sorghum aphid.

**Abstract:**

Appropriate reference genes must be selected for accurate qRT-PCR data to conduct a thorough gene expression analysis in the sorghum aphid (*Melanaphis sacchari*, Hemiptera, Aphididae). This approach will establish a foundation for gene expression analysis and determines the efficacy of RNA interference in the sorghum aphid. Nine potential reference genes, including *Actin*, *18S*, *GAPDH*, *RPL7*, *EF-1α*, *EF-1β*, *28S*, *HSP70*, and *TATA*, were assessed under various experimental conditions to gauge their suitability based on qRT-PCR Ct values. The stability of these candidate reference genes in diverse experimental setups was analyzed employing several techniques, including the ΔCt comparative method, geNorm, Normfinder, BestKeeper, and RefFinder. The findings revealed that the quantity of ideal reference genes ascertained by the geNorm method for diverse experimental conditions remained consistent. For the developmental stages of the sorghum aphid, *RPL7* and *18S* proved to be the most dependable reference genes, whereas *GAPDH* and *EF-1β* were recommended as the most stable reference genes for different tissues. In experiments involving wing dimorphism, *EF-1α* and *GAPDH* were identified as the optimal reference gene pair. Under varying temperatures, *EF-1α* and *EF-1β* were found to be the most dependable gene pair. For studies focusing on insecticide susceptibility, *18S* and *TATA* emerged as the most stable candidate reference genes. Across all experimental conditions, *EF-1α* and *EF-1β* was the optimal combination of reference genes in the sorghum aphid. This research has pinpointed stable reference genes that can be utilized across various treatments, thereby enhancing gene expression studies and functional genomics research on the sorghum aphid.

## 1. Introduction

*Melanaphis sacchari* is a harmful sorghum aphid (Hemiptera, Aphididae) and a major pest causing damage to forage crops such as sugarcane, wheat, corn, Sudan grass, peach, and others [1,2,3,4,5]. The damage results from the direct feeding and transmission of many plant viruses [6,7,8]. Additionally, the honeydew secreted by the aphid causes mildew disease during later growth stages, reducing forage crop yield and affecting quality and flavor [9]. Quantifying gene expression through real-time fluorescence quantitative polymerase chain reaction (qRT-PCR) analysis is dependable and reproducible [10,11,12]. This technique is valuable for assessing insect gene function and is effective in verifying gene expression [13,14]. However, difficulties persist in standardizing qRT-PCR data due to inconsistencies in sample amounts, RNA extraction methods, reverse transcription processes, and PCR amplification efficiency. Standard reference genes are employed to guarantee the accuracy of target gene expression results obtained from qRT-PCR analyses [14,15,16]. The ideal reference gene is one with consistent expression under all experimental conditions [17]. Research has demonstrated that reference gene expression stability is relative, and no single reference gene exhibits consistent expression under all experimental conditions, in all tissues, and across different species [18,19,20,21].

The consistent expression of a reference gene signifies stable expression within a specified range [22,23,24,25]. *Actin* (beta-actin gene) [26], *18S ribosomal* (*18S rRNA*) [27], *28S ribosomal* (*28S rRNA*) [28], *glyceraldehyde-3-phosphate dehydrogenase* (*GAPDH*) [29], *alpha-tubulin* (*α-TUB*) [30], *TATA-box-binding protein* (*TATA*) [31], *ribosomal protein L7* (*RPL7*) [32], *elongation factor 1 alpha* (*EF-1α*) [33], *elongation factor 1 beta* (*EF-1β*) [34], *ribosomal protein L18 (RPL18*) [35], and various other genes are commonly used as reference genes in insects. Screening reference genes is crucial under specific experimental conditions, as some reference genes may not have consistent expression levels across all conditions [20]. To validate the chosen reference genes, the expression levels of *Heat Shock Protein 70 (HSP70)*, a critical gene involved in the stress response, was utilized [36]. Researchers use different reference genes to study gene expression in different insect species. For example, reference genes have been described for model insects, such as *Drosophila melanogaster*, *Nilaparvata lugens (Stal)*, and *Tribolium castaneum* [37,38,39], while reference genes for the sorghum aphid have not been documented. The stability of potential sorghum aphid genes will be analyzed in this study by measuring the Ct values of eight frequently used normalization genes (*Actin*, *18S*, *GAPDH*, *RPL7*, *EF-1α*, *EF-1β*, *28S*, and *TATA*) under various treatments. Consequently, the reference genes for the sorghum aphid under various treatments will be identified, laying the groundwork for subsequent studies of sorghum aphid gene function.

## 2. Materials and Methods

### 2.1. Insect Rearing

The sorghum aphids were collected from plantations at Anhui Science and Technology University, Fengyang County, Anhui Province, China. They were later raised on sweet sorghum seedlings in a controlled environment, maintaining a temperature of 24 ± 1 °C, relative humidity of 60 ± 10%, and a photoperiod of 14L: 10D. This setup allowed for continuous feeding over multiple generations. The developmental stages of the sorghum aphids encompassed five instars (1st, 2nd, 3rd, 4th instar, and adulthood), with the larval stage lasting 6–7 days and the overall lifespan of the sorghum aphid spanning 30–35 days.

### 2.2. Biotic Conditions

#### 2.2.1. Developmental Stage

The sorghum aphid undergoes five stages of development, including first, second, third, fourth, and adult. The five instars of sorghum aphids were collected, and female aphids were transplanted onto sorghum leaves in vitro. After 12 h, the female aphids were removed, leaving behind the hatched nymphs. Five hundred first-instar and second-instar nymphs, as well as three hundred third-instar, fourth-instar, and adult aphids, were collected for the experiment. All of the aphids were subjected to three biological replications. Afterward, the specimens were stored at −80 degrees Celsius [10].

#### 2.2.2. Tissue and Wing Dimorphism

Three distinct body regions, namely the head, legs, thorax, and abdomen, were isolated. Three hundred wingless aphids were dissected using tweezers under a stereomicroscope, with each sample being collected three times and stored as previously mentioned. Furthermore, fifty mature insects, both winged and wingless, were each gathered and placed in separate 1.5 mL centrifuge tubes. Each sample used in the study was duplicated three times and stored according to the previous instructions [17].

### 2.3. Abiotic Conditions

#### 2.3.1. Temperature Treatment

The sorghum aphids were subjected to different temperatures (10, 18, 24, 30, and 35 °C) for 2 h. The experimental temperatures ranged from 10 to 35 °C. Each temperature treatment involved the repeated exposure of 50 aphids, with each sample collected three times and stored according to established protocols.

#### 2.3.2. Insecticide Treatment

Initially imidacloprid was dissolved in acetone and then diluted with distilled water to achieve an LC concentration of 500.25 mg L^−1^. The sorghum seedlings infested with aphids were immersed in an imidacloprid solution for 10 s, while the control group of aphids were immersed in distilled water for the same duration. Following the treatment, the surviving sorghum aphids were selected after 24 h of pesticide treatment. The surviving aphids treated with imidacloprid were selected and control aphids were collected. All samples utilized in the experiment were replicated three times and stored per established procedures.

### 2.4. Total RNA Extraction and cDNA Synthesis

Total RNA was isolated from the *M. sacchari* sample with Trizol reagent (Tian gen Biochemical Technology, Beijing, China), followed by genomic DNA removal using DNaseΙ treatment (TAkara Bio, Beijing, China). RNA quantification was conducted using the Nanodrop 2000 spectrophotometer (Thermo Scientific, Waltham, MA, USA). The first-strand complementary DNA (cDNA) was synthesized using the PrimeScriptR RT reagent kit and stored at −20 °C for further use.

### 2.5. Selection of Reference Genes and Design of the Primers

The nucleotide gene sequences of the sorghum aphid were obtained from the NCBI database (https://www.ncbi.nlm.nih.gov/datasets/genome/; accessed on 22 May 2023), with candidate reference genes primarily composed of *Actin* [26], *18S* [27], *GAPDH* [29], *RPL7* [32], *EF-1* [33], *EF-1*β [34], *28S* [28], *TATA* [31], and the selected target gene *HSP70* [36]. Primers designed for the nine genes were constructed using Primer Premier 5.0, followed by traditional PCR cloning of the candidate genes and subsequent comparison and verification of the genetic sequences. The procedure involved creating primers to clone and identify potential reference genes for the sorghum aphid (Table 1).

### 2.6. Real-Time qPCR Analysis

The SYBR qPCR Master Mix from Vazyme Biotech was utilized for the real-time qPCR analysis, with the expression of candidate reference genes evaluated using the ViiA 7 system (ABI, Foster City, CA, USA). qRT-PCR reactions were performed according to the manufacturer’s guidance. The thermal cycling protocol included an initial denaturation at 95 °C for 30 s, followed by 40 cycles of denaturation at 95 °C for 10 s and annealing/extension at 60 °C for 30 s. After cycling, the melting curves were generated by increasing the temperature from 60 to 95 °C at a rate of 0.2 °C per second to dissociate double-stranded DNA. Relative standard curves were created by performing five-fold serial dilutions of cDNA (diluted 1, 5, 25, 125, and 625 times). The efficiency of amplification for the target genes was calculated by the formula E = 10^(−1/Slope)^ − 1. Each sample was analyzed in three biological replicates, and each qRT-PCR reaction was conducted in triplicate [40].

#### 2.6.1. Analysis of Expression Stability of Candidate Reference Genes

The Ct values derived from qRT-PCR were examined using several approaches, including the ΔCt method [15], geNorm (version v3.4) [41], Normfinder (version 20) [42], and BestKeeper (version 20) [43]. The reference genes were assessed with geNorm and NormFinder by converting the Ct values into 2^ΔCt^ for further analysis. geNorm determines the V_n_/V_n+1_ pairwise variation, which reveals the difference between consecutive normalization factors, thereby identifying the optimal number of reference genes required for precise normalization. The Ct values were evaluated directly using the ΔCt and BestKeeper methods. The comprehensive final ranking was achieved using RefFinder. RefFinder is an extensive online tool designed to appraise and filter reference genes from extensive experimental datasets (https://blooge.cn/RefFinder/?type=reference; accessed on 22 January 2024). This software incorporates key computational tools, including geNorm, NormFinder, BestKeeper, and the comparative ΔCt method, to assess and rank potential reference genes. RefFinder utilizes the rankings generated by each tool to assign weights to the genes, then determining the final ranking by calculating the geometric mean of these weights.

#### 2.6.2. Verification of the Stability of Selected Reference Genes

To validate the selected reference genes, the expression levels of *HSP70*, a key stress-responsive gene, were analyzed. *HSP70* expression was assessed across different developmental stages, temperature variations, and insecticide treatments. The expression level of the *HSP70* gene was determined using the 2^−ΔΔCt^ method [44]. Statistical analysis of target gene expression levels was conducted using one-way ANOVA with SPSS Statistics 17.0, setting the significance threshold at *p* < 0.05.

## 3. Results

### 3.1. Cloning of the Candidate Genes and qRT-PCR Primer Selection

Primer sequences for the reference genes of the candidates were created using the gene sequence information from the NCBI database (Table 2). The potential genes were analyzed using RT-PCR. Subsequently, all amplicons were sequenced and were identical to their respective transcripts (see Appendix A for details). Furthermore, every gene candidate was confirmed to be a sole amplicon of the accurate size through the utilization of 2% agarose gel electrophoresis. Fluorescent primers for qPCR were created based on sequence information, and the accuracy of the eight reference genes as well as the heat shock protein gene was verified for qRT-PCR. This validation was achieved by detecting a single peak in the melting curve analysis and observing distinct bands on an agarose gel. This validation was accomplished by identifying a sole peak in the melting curve analysis and observing clear bands on an agarose gel. A standard curve was generated using the Ct value on the y-axis and the logarithm of the relative copy number on the x-axis to evaluate the linearity of this correlation by determining. the slope of the linear equation and the regression coefficient (R^2^) [45]. The amplification efficiency of the potential reference genes varied between 101.577 and 108.547, while the correlation coefficients varied between 0.991 and 1.000. The primers for the potential reference genes and the gene encoding HSP were chosen.

### 3.2. Evaluation of the Levels of Gene Expression in Potential Reference Genes

The quantity of potential reference genes available is a key factor in the screening process [23]. The sorghum aphid exhibited Ct values between 15.80 and 27.967 for the eight potential reference genes, satisfying the required standards for gene expression as reference genes [12]. The *EF-1α* gene exhibited the smallest Ct value, suggesting the highest expression level, whereas the *TATA* gene displayed the largest Ct value, indicating the lowest expression level. Specifically, the expression levels of the *EF-1α*, *EF-1β*, *GAPDH*, and *RPL7* genes were higher, whereas the gene expression levels of the *Actin*, *TATA*, *28S*, and *18S* genes were lower. These results demonstrate that the Ct values of the *EF-1α*, *GAPDH, RPL7*, and *EF-1*β genes were smaller than those of *Actin* and *18S*, whereas the Ct values of *TATA* and *28S* were larger (Figure 1).

### 3.3. Stability of the Reference Genes under Various Experimental Conditions

#### 3.3.1. Development Stages

Four software programs, ΔCt, BestKeeper, NormFinder, and geNorm, were used to evaluate the stability ranking of potential reference genes across different developmental stages of the sorghum aphid. The *RPL7* and *EF-1β* genes emerged as the most stable (Table 3 and Table 4). A detailed stability ranking of the eight candidate reference genes was generated through RefFinder analysis. The comprehensive stability ranking ordered the candidate reference genes as follows: *RPL7*, *EF-1β*, *18S*, *EF-1α*, *GAPDH*, *TATA*, *Actin*, and *28S* (Figure 2A), with 28S being the least stable. Additionally, the V2/3 value from geNorm software was <0.15 (Figure 3). Therefore, for quantifying gene expression levels across various developmental stages in the sorghum aphid, two genes, *RPL7* and *EF-1β*, were selected as reference genes.

#### 3.3.2. Different Tissue

According to the ΔCt and NormFinder evaluations (Table 3), the *GAPDH*, *EF-1β*, and *TATA* genes were the most reliable. The RefFinder results indicated that the *GAPDH* and *EF-1β* genes exhibited the highest stability, ranking in the order of stability as *GAPDH*, *EF-1β*, *TATA*, *Actin*, *RPL7*, *18S*, *EF-1α*, and *28S* (Figure 2B). Furthermore, the 28S gene showed the least stability. According to the geNorm analysis, the V2/3 pairwise variation value fell below the recommended threshold of 0.15 (Figure 3). Therefore, for effective normalization in this study, the use of *GAPDH* and *EF-1β* genes was essential (Table 4).

#### 3.3.3. Wing Dimorphism

Based on analyses from ΔCt, geNorm, and NormFinder, *EF-1α* and *GAPDH* emerged as the most dependable genes (Table 3). Additionally, RefFinder analysis confirmed the robustness of *EF-1α* and *GAPDH*, ranking their stability as *EF-1α* > *GAPDH* > *Actin* > *18S* > *EF-1β* > *TATA* > *RPL7* > *28S* (Figure 2C). Both the analytical software and RefFinder consistently revealed that the *28S* gene had the lowest stability. Additionally, the geNorm analysis demonstrated that the V2/3 ratio was <0.15 (Figure 3), suggesting that two candidate reference genes would suffice for accurate normalization across various dimorphisms. Consequently, *EF-1α* and *GAPDH* were crucial for appropriate normalization in this scenario (Table 4).

#### 3.3.4. Insecticide Treatment

Sorghum aphids treated with imidacloprid were compared with untreated aphids used as a control group. Candidate reference genes for assessing pesticide effects on sorghum aphids were identified. Analysis utilizing ΔCt, NormFinder, and additional software programs indicated that the *TATA* gene exhibited the greatest stability, whereas BestKeeper and geNorm recognized *18S* as the most stable gene (Table 3 and Table 4). Gene stability was ranked by RefFinder in the following order from most stable to least stable as *18S*, *TATA*, *EF-1β*, *Actin*, *EF-1α*, *RPL7*, *GAPDH*, and *28S* (Figure 2D). Additionally, geNorm analysis underscored the importance of selecting two reference genes for accurate gene expression analysis in studies involving pesticide effects on sorghum aphids (Figure 3).

#### 3.3.5. Temperature

The temperature conditions were set to 10, 18, 24, 30, and 35 °C, and third or fourth-instar aphids were exposed to these temperatures for 2 h in an artificial climate box. According to NormFinder and geNorm analyses, *EF-1α* showed the highest stability among the genes, while the ΔCt method identified *EF-1β* as the most reliable (Table 3). RefFinder’s assessment further detailed the stability ranking from most to least stable: *EF-1α*, *EF-1β*, *GAPDH*, *TATA*, *18S*, *RPL7*, *Actin*, and *28S* (Figure 2E). Consistently, the *28S* gene was the most unstable gene at all temperatures and in every analysis. Additionally, according to the geNorm assessment, because the V2/3 ratio was <0.15 (Figure 3), It is advisable to utilize EF-1α and EF-1β for appropriate standardization within this specific group (Table 4).

#### 3.3.6. RefFinder’s Comprehensive Ranking of Reference Genes

RefFinder determined that *EF-1β* and *EF-1α* were the most dependable genes (Table 4), with *EF-1β* being the most stable followed by *EF-1β*, *EF-1α*, *GAPDH*, *18S*, *TATA*, *RPL7*, *Actin*, and *28S* (Figure 2F). The geNorm program was utilized to assess the pairwise differences (V_n_/V_n+1_) among normalization factors NF_n_ and NF_n+1_, identifying the ideal quantity of reference genes needed for accurate normalizationA value below 0.15 indicates that adding more reference genes would not significantly enhance normalization.

### 3.4. Selection Verification of Reference Genes

The *HSP70* gene served as a validation marker to assess the stability of reference genes under various temperature conditions. RefFinder analysis identified *EF-1α* and *EF-1β* as suitable reference genes for studying *HSP70* expression across different temperatures (Figure 4A). Interestingly, no significant differences in *HSP70* expression were observed at lower temperatures (10 and 18 °C) compared to moderate temperature (24 °C) (*p* > 0.05). Similarly, there were no notable variations in *HSP70* activity between 30 °C and 24 °C (*p* > 0.05). However, *HSP70* expression was significantly higher in sorghum aphids at 35 °C than under other temperature conditions (*p* < 0.05).

Moreover, *18S* and *TATA* were chosen as reference genes for the pesticide treatment assessment based on the RefFinder analysis. The level of *HSP70* gene activity in sorghum aphids was established. The results indicate a significant increase in *HSP70* levels in sorghum aphids that were exposed to imidacloprid compared to the control group (*p* < 0.05) (Figure 4B). *HSP70* gene expression in aphids exposed to the pesticide increased 54.86-fold compared to the control group, demonstrating a significant physiological reaction from the sorghum aphids to the stress induced by the pesticide.

## 4. Discussion

Analysis of gene expression is paramount in contemporary biology and represents a central focus of ongoing research [45,46,47,48]. Such analyses are instrumental in comprehending the molecular mechanisms governing various processes related to insect growth and development [49]. qRT-PCR is the premier technique for measuring gene expression because of its exceptional sensitivity and efficiency [48,50]. It has been widely employed to measure mRNA expression levels under various experimental settings, as well as to evaluate the effectiveness of RNA interference [51]. It is important to recognize that the activity of certain target genes may be affected by control genes in a particular experimental context. Consequently, in biological experiments involving target insects, the use of distinct internal reference genes is imperative to ensure accurate analysis of target gene expression [52]. Various insect species, such as *Tribolium castaneum* [53], *Apis mellifera* [54], *Nilaparvata lugens* [55], and the cotton aphid [56], have had reference genes discovered. However, a lack of data exists on the suitable reference genes for the sorghum aphid, a pest that mainly affects forage crops like sorghum, sorghum Sudanense, and Sudan grass [57]. The stability of eight candidate reference genes was assessed for the sorghum aphid under different experimental conditions in this study.

Consistently, the V2/3 value stayed below 0.15 in every instance, as determined by geNorm software, suggesting that two reference genes were the best choice for all experiments. The geNorm software identified the best number of control genes for greenbug (*Schizaphis graminum* Rondani) and Aphidius gifuensis (*Hymenoptera Aphidiidae*). Notably, the ideal number of control genes for sorghum aphids matched those for *S. graminum* and *A. gifuensis* [10,58]. *28S* and *α-TUB* were the best reference genes in *S. graminum*, while 28S was the least stable reference gene in sorghum aphid in different treatments [10]. A ccording to the RefFinder analysis, *RPL7* and *EF-1β* were identified as the most dependable reference genes throughout different phases of sorghum aphids, with *RPL7* also emerging as the preferred option for cotton aphids. Notably, the reference genes for cotton aphids were similar to those identified for sorghum aphids at different developmental stages [56]. *GAPDH* and *EF-1β* are the most stable genes across different tissue types, as supported by the ΔCt, BestKeeper, NormFinder, geNorm, and RefFinder analyses. *EF-1α* was consistently identified as a dependable control gene by NormFinder, geNorm, and RefFinder in relation to differences in wing morphology and temperature, while *18S* and *TATA* showed the highest stability in response to insecticide exposure. RefFinder’s stability assessment determined that *EF-1β* and *EF-1α* are the most dependable genes in regards to stability. In contrast, the *28S* gene was consistently unstable across various software analyses and was deemed unreliable as a reference gene for cotton aphids.

Molecular chaperones, known as *Heat shock proteins* (HSPs), are quickly produced when cells encounter a range of environmental stress conditions, including heat shock, cold exposure, insecticide exposure, heavy metal contamination, and diapause [59]. We analyzed the levels of *HSP70* expression under various temperature and insecticide treatment scenarios to confirm the chosen reference genes. *EF-1α* and *EF-1β* were selected as the reference genes for the different temperature conditions during the evaluation of *HSP70* expression. The findings reveal significantly higher *HSP70* expression at 35 °C than at the other temperatures, aligning with previous research demonstrating the low tolerance of the sorghum aphid to high temperatures; 24 °C is the optimal temperature for sorghum aphids. Consequently, the substantial increase in *HSP70* expression at 35 °C was a response to cope with the high-temperature environment. In addition to the environmental temperature, insecticide stress is a significant factor affecting insect responses. Non-lethal levels of beta-cypermethrin increase the expression levels of Rp*HSP70*-1 and Rp*HSP70*-2 in *Rhopalosiphum padi* [60]. After the insecticide treatment, *HSP70* expression was examined using *18S* and *TATA* as reference genes in sorghum aphids. These findings show that *HSP70* gene expression increased by 54.86 times after exposure to the pesticide compared to the control group, demonstrating a physiological reaction of the sorghum aphid to stress caused by the pesticide.

## 5. Conclusions

The stability of nine genes used as reference were thoroughly assessed under various experimental conditions using five different methods. Consistently, the geNorm method determined the best number of reference genes for every experiment. During the course of development, *RPL7* and *EF-1β* demonstrated strong reliability as reference genes. In various tissues, *GAPDH* and *EF-1β* exhibited stable expression. *EF-1α* and *GAPDH* were identified as the most effective reference gene pair for experiments related to wing dimorphism. *EF-1α* and *EF-1β* were found to be the most dependable gene pair under varying temperatures, with *18S* and *TATA* consistently showing positive responses to insecticide exposure. *EF-1α* and *EF-1β* emerged as the optimal pair of reference genes for aphids in various experimental settings. This extensive research successfully pinpointed genes that remained constant under various treatment conditions, providing a strong foundation for future investigations into gene expression and functional genomics in the sorghum aphid.

## Figures and Tables

**Figure 1 insects-15-00522-f001:**
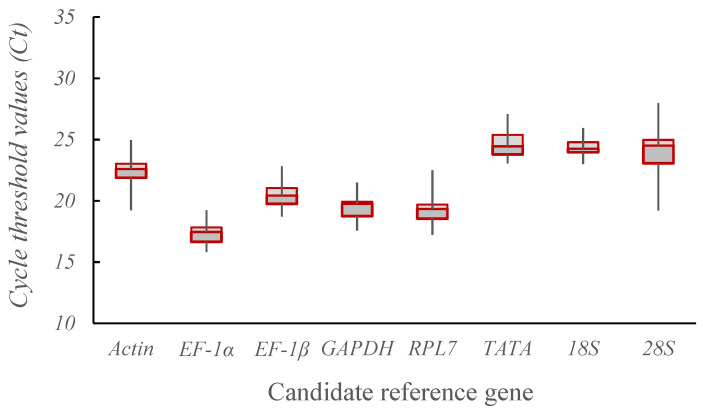
Expression levels of candidate reference genes of *Melanaphis sacchari* indicated by Ct values.

**Figure 2 insects-15-00522-f002:**
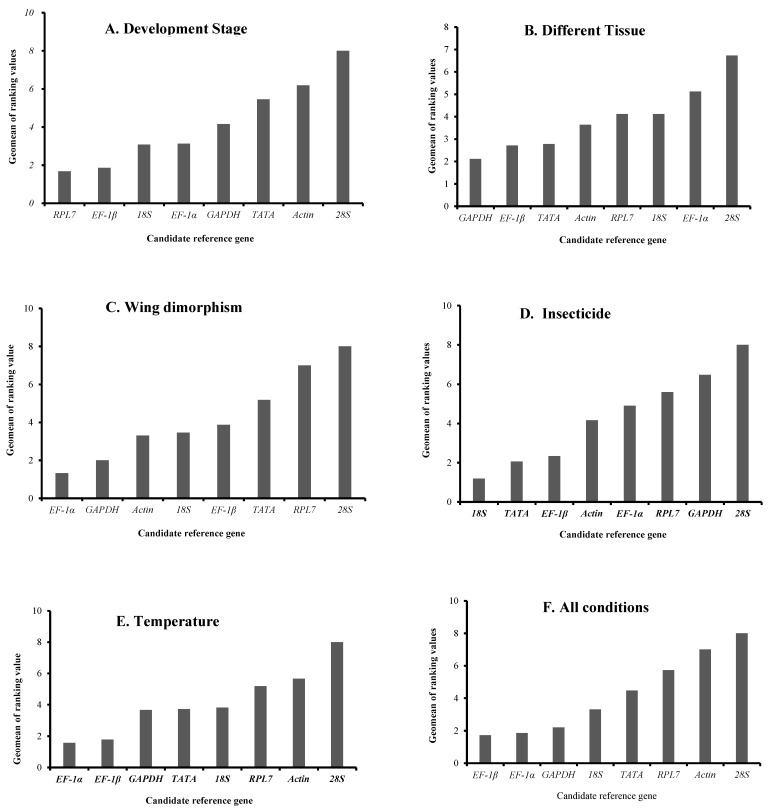
Expression stabilities of the candidate reference genes under different experimental conditions calculated by RefFinder. (**A**) Different developmental stages. (**B**) Different tissues. (**C**) Wing dimorphism. (**D**) Insecticide treatments. (**E**) Temperature. (**F**) Pooled samples. A lower geomean of ranking value indicates more stable expression.

**Figure 3 insects-15-00522-f003:**
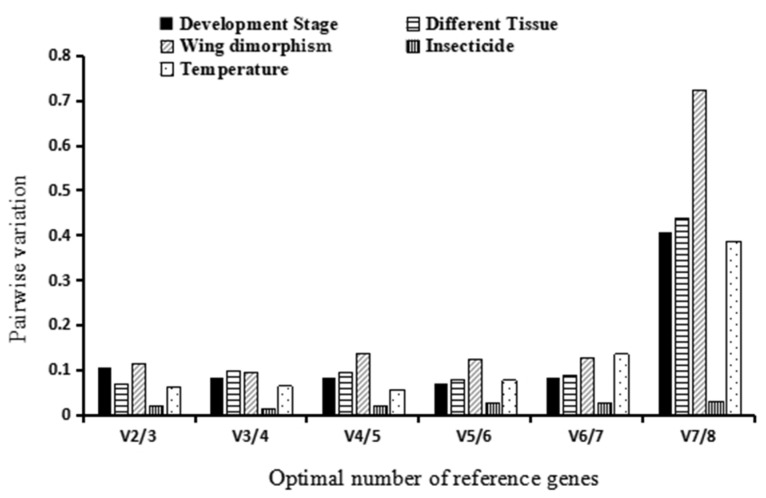
Optimal numbers of reference genes for normalization in *Melanaphis sacchari*.

**Figure 4 insects-15-00522-f004:**
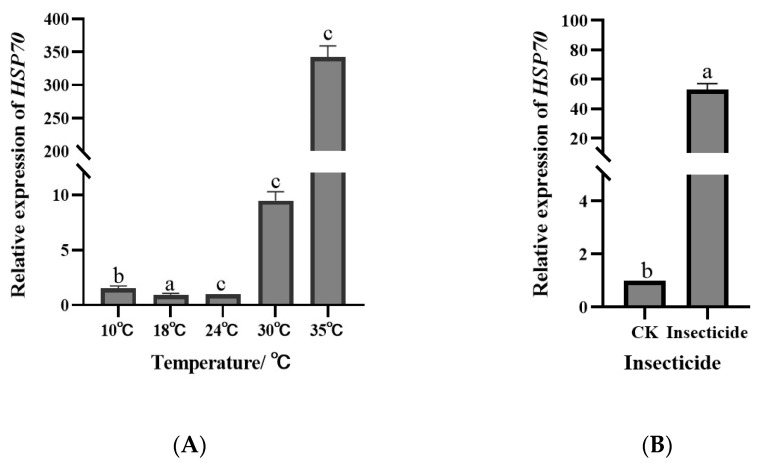
Relative expression levels of an *HSP70* target gene calculated using various sets of reference genes. (**A**) Expressions of *HSP70* of sorghum aphid for different temperatures. (**B**) Expressions of *HSP70* of sorghum aphid in insecticide treatment. Different lowercase letters represent differences in expression.

**Table 1 insects-15-00522-t001:** The primers for cloning and identification of candidate reference genes and one heat shock protein gene *HSP70* of sorghum aphid *Melanaphis sacchari*.

Genes	Primer Sequences (5′-3′)	Gene	Accession No.	Length (bp)
*Actin*	F:TGCTGTCTTCCCGTCCAT	*Actin*	XM_025350371.1	746
	R:TTTCGTGGATACCGCAAG			
*18S*	F:TGGCAGTGTTGCTGAAGA	*18S ribosomal*	XM_025343011.1	624
	R:CCACTCCTTGATCGTCCT			
*GAPDH*	F:GTAGCCATCAATGACCCA	*Glyceraldehyde-3-phosphate*	XM_025343821	700
	R:GGCAGCACCTCTACCATC			
*RPL7*	F:ACGTAAGGCTCGTACAGC	*Ribosomal protein L 7*	XM_025345708.1	406
	R:TGAAAGGCCAAAGGAAGT			
*EF-1α*	F:TCATTGACGCACCTGGAC	*Elongation factor 1 alpha*	XM_025346188.1	574
	R:AAGACCACAACCATACCG			
*EF-1*β	F:ATGGCTGCCGTTGACTTA	*Elongation fator 1 beta*	XM_025337882.1	644
	R:TTGAATGCAGCAATGTCC			
*28S*	F:AAAAGGTCCTGGACGAAA	*28S ribosomal*	XM_025343486.1	1136
	R:ACGCCATAACAGGTAACATAC			
*TATA*	F:GGATCAAATGTTACCGAGTC	*TATA box binding protein*	XM_025347315.1	828
	R:CGTACTTTGGCACCTGTC			
*HSP70*	F:ATGGTCGGAAAGACTGCTAT	*Heat Shock Protein 70*	XM_025350978.1	1004
	R:ATGTCGTGGATGTCTCCCT			

**Table 2 insects-15-00522-t002:** The amplification efficiencies of qPCR primers of candidate reference genes and *HSP 70* gene in sorghum aphid *Melanaphis sacchari*.

Genes	Primer Sequences (5′-3′)	Accession No.	Length (bp)	Amplification Efficiency (%)	R^2^
*Actin*	F:TTTGGACTCAGGTGACGGTG	XM_025350371.1	166	105.797	0.9997
	R:TTCACGCTCAGCAGTAGTGG				
*18S*	F:ACATTGGTGATGGCGTTCCA	XM_025343011.1	172	104.623	0.9993
	R:AAGACTGCTCTAGCGTTGCG				
*GAPDH*	F:AGGTGTTCTCTGAACGCGAC	XM_025343821	78	105.681	0.9990
	R:CACCGGTGGATTCAACAACG				
*RPL7*	F:AACGCGCTGAAGCTTATGTT	XM_025345708.1	151	105.575	0.9997
	R:GCCACTTGATTCACACCACG				
*EF-1α*	F:CAACTGACAAGGCTCTCCGT	XM_025346188.1	77	101.577	0.9992
	R:ACGACCAACTGGGACTGTTC				
*EF-1*β	F:TTTGCTGCGATTCAAGCACC	XM_025337882.1	141	103.375	0.9990
	R:TGCAGCATTAGCCGAAGACA				
*28S*	F:GCTTGAGCAAGGTCACGTCT	XM_025343486.1	160	102.747	0.9997
	R:CCAGTGAAACGCCTAGACCA				
*TATA*	F:GGTATGCACTGGCGCAAAAA	XM_025347315.1	167	108.547	0.9995
	R:GTACCAATCCCTCAAGCCGT				
*HSP70*	F:CTGCGGAAAGCCCAAAATCC	XM_025350978.1	119	109.320	0.9943
	R:TTCGTCAGCACCATCGAACT				

**Table 3 insects-15-00522-t003:** Expression stabilities of the candidate reference genes in *Melanaphis sacchari* under different experimental conditions by ΔCt, BestKeeper, NormFinder, and geNorm.

Treatments	Genes	ΔCt		BestKeeper		NormFinder		geNorm	
		Stability	Rank	Stability	Rank	Stability	Rank	Stability	Rank
Development Stages	*Actin*	0.962	7	0.57	2	0.311	6	0.481	6
	*18S*	0.827	5	0.45	1	0.292	5	0.418	5
	*GAPDH*	0.817	4	0.72	5	0.107	1	0.390	4
	*RPL7*	0.772	1	0.58	3	0.271	4	0.321	2
	*EF-1α*	0.784	3	0.81	6	0.123	2	0.283	1
	*EF-1β*	0.777	2	0.67	4	0.158	3	0.283	1
	*28S*	3.029	8	1.94	8	2.251	8	1.177	7
	*TATA*	0.869	6	0.91	7	0.408	7	0.351	3
Different Tissues	*Actin*	0.340	4	0.36	7	0.319	4	0.011	1
	*18S*	0.350	5	0.37	8	0.332	5	0.011	1
	*GAPDH*	0.260	1	0.20	5	0.042	1	0.126	3
	*RPL7*	0.310	3	0.32	6	0.253	3	0.043	2
	*EF-1α*	0.390	6	0.08	2	0.352	6	0.289	6
	*EF-1β*	0.300	1	0.04	1	0.145	2	0.227	5
	*28S*	0.47	7	0.14	4	0.465	7	0.335	7
	*TATA*	0.26	2	0.14	3	0.042	1	0.172	4
Wing dimorphism	*Actin*	1.059	4	1.15	2	0.068	4	0.455	4
	*18S*	1.141	6	1.01	1	0.068	3	0.552	5
	*GAPDH*	0.954	2	1.48	4	0.001	2	0.004	1
	*RPL7*	1.312	7	2.07	7	1.023	7	0.646	6
	*EF-1α*	0.952	1	1.48	3	0.001	1	0.004	1
	*EF-1β*	1.021	3	1.72	5	0.524	5	0.228	2
	*28S*	4.482	8	2.56	8	4.008	8	1.931	7
	*TATA*	1.089	5	1.83	6	0.693	6	0.303	3
Insecticide	*Actin*	0.194	5	0.06	4	0.094	6	0.044	2
	*18S*	0.259	3	0.01	1	0.032	2	0.016	1
	*GAPDH*	0.235	7	0.12	7	0.094	5	0.124	6
	*RPL7*	0.205	6	0.07	5	0.113	7	0.052	3
	*EF-1α*	0.176	2	0.09	6	0.052	4	0.101	5
	*EF-1β*	0.169	4	0.03	3	0.020	1	0.071	4
	*28S*	0.240	8	0.18	8	0.163	8	0.153	7
	*TATA*	0.199	1	0.02	2	0.049	3	0.016	1
Temperature	*Actin*	0.801	6	0.65	7	0.301	6	0.329	5
	*18S*	0.954	7	0.27	1	0.163	5	0.221	3
	*GAPDH*	0.672	4	0.35	4	0.144	4	0.138	1
	*RPL7*	0.653	3	0.49	6	0.086	3	0.256	4
	*EF-1α*	0.632	2	0.33	3	0.048	1	0.138	1
	*EF-1β*	0.626	1	0.32	2	0.051	2	0.175	2
	*28S*	2.834	8	2.23	8	2.140	8	1.159	7
	*TATA*	0.673	5	0.39	5	0.727	7	0.507	6

**Table 4 insects-15-00522-t004:** Recommended reference genes in *Melanaphis sacchari* for the different experimental conditions.

Biotic Factor	Reference Gene	Abiotic Factor	Reference Gene
Developmental stage	*RPL7*, *EF-1β*	Temperature	*EF-1α*, *EF-1β*
Tissue	*GAPDH*, *EF-1β*	Insecticide	*18S*, *TATA*
Wing dimorphism	*EF-1α*, *GAPDH*	All conditions	*EF-1β*, *EF-1α*

## Data Availability

The data presented in this study are available on request from the corresponding author.

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
