# Peer review of "Identification and Evaluation of qRT-PCR Reference Genes in Melanaphis sacchari"

_insects, 2024, doi:10.3390/insects15070522_

Round 1
Reviewer 1 Report (Previous Reviewer 1)
Comments and Suggestions for Authors
After previous revisions the manuscript is substantially improved and nearly ready for publication. A few minor concerns should be addressed to improve the overall quality of the manuscript, as follows:
Abstract.
Line 34. “Moreover, different temperatures” should be deleted.
Introduction
Line 68-69. “Different insect species use different reference genes.”
This is incorrect. Insects don’t use reference genes. It would be more correct to say that “Researchers use different reference genes to study gene expression in different insect species.”
Line 69 (also line 317). Change “discovered” to “described”
Line 71. Change “the reference gene” to “reference genes”
Materials and Methods
Line 110 (and elsewhere). “IMIDACLOPRID”
It is not clear why the pesticide, Imidacloprid, is written in all capital letters. Is this a specific product from a specific company? If so, then at first mention, the company and its headquarters/location should be mentioned in parenthesis.
Line 114. “The surviving sorghum aphid was selected”
Was there only one surviving aphid? If not, this should be written as “surviving sorghum aphids were selected”
Line 116-117. Capitalize “all” and remove the period after “times”
Line 120. Change “extracted” to “eliminated”
Line 172-73. There seems to be a smaller font size used here. Please adjust for consistent formatting throughout the manuscript.
Results
Line 196-197. “meeting the necessary criteria for gene expression as reference genes”
Necessary according to who? Is this established by MIQE standards or some other publication. The way this statement is worded, it seems like there should be a citation here.
Line 209. Delete “such as” the statement stands on its own, and is more accurate, without this phrase.
Line 246. “Genorm” should be geNorm” Please ensure this term is written correctly and consistently throughout the manuscript.
Line 329-330. Remove italics from all text after “S. graninum”
References.
Reference 12. MIQE guidelines. The reference is wrong. The authors are written in reverse order. Bustin is the first author and Wittwer is the last author. Please ensure all references are formatted correctly.
Comments on the Quality of English Language
The manuscript is very well written. A few minor suggestions have been made to the authors, otherwise I have no other concerns.
Author Response
Please see the attachment.

Reviewer 2 Report (Previous Reviewer 2)
Comments and Suggestions for Authors
The quality of the manuscript has greatly improved compared to the previous version. This work provides important information for the selection of reference genes in quantitative PCR. I think the manuscript can currently be accepted for publication.
Author Response
Please see the attachment.

This manuscript is a resubmission of an earlier submission. The following is a list of the peer review reports and author responses from that submission.
Round 1
Reviewer 1 Report
Comments and Suggestions for Authors
The authors report on characterization of candidate reference genes for qRT-PCR analysis in the sorghum aphid, Melanaphis sacchari. This is an important foundational study for future molecular analyses in this insect. Identification of the most stably expressed genes are considered under different experimental conditions including across different life stages, tissue and temperatures, and also subsequent to pesticide exposure. Finally, validation of the most stable candidate reference gene is undertaken by examining expression of a heat shock protein across different life stages, at different temperatures, and after pesticide exposure.
The manuscript is poorly written, both in terms of language, but also in clarity of the message, throughout the document. Specific examples where clarity is needed are given below. And it is advised that after all revisions are made, that the manuscript be thoroughly language edited before re-submission. Furthermore, important information is missing, or not clear, in several sections of the Materials and Methods. This makes it difficult to properly evaluate the Results and Discussion of the manuscript until these details are provided; specific examples are given below. Thus, it can only be recommended that Major Revisions are required for this manuscript before it may be deemed suitable for publication.
Specific comments for each section are given as follows:
Introduction
Line 50. “gene quantification” should be “gene expression quantification”
Materials and Methods.
Line 93 (and elsewhere in other sub-sections). “every sample was repeatedly collected three times”
It is not clear what this means. Do you mean there were 3 biological replicates sampled for each condition?
Section 2.4 Total RNA extraction and cDNA synthesis.
For this procedure, was there done a genomic DNA degradation step? If this was done or not done, it should be mentioned, since genomic DNA can be a source of contamination for qPCR assays. This is especially important, because in section 2.6, there is no mention whether any negative controls were performed in the qRT-PCR assays.
Line 122. “According to the transcriptome data of Sorghum Aphid”
What transcriptomic data? A citation/reference is needed here for this dataset.
Line 124-125. “and the genetic sequence was compared and verified.”
The reference data and results of new sequencing should be included in this report, the sequence data and/or alignment should be included in this report, maybe as supplemental data.
Section 2.6. Real-time qPCR analysis.
Line 134, primers are mentioned, but there are no references to the primers in this section. This info is presented in the results section as Table 2, but that reference should first be made here in this section because that is the first mention of the primers.
Line 136-137. “The amplification efficiency of the target genes were estimated”.
How was this done? Does this suggest a dilution series was utilized? Which dilution series? What was diluted? cDNA or the cloned plasmids? Greater clarity is needed here.
Section 2.6.1. Analysis of expression stability of candidate reference gene.
Throughout this section, for each of the software programs that were used, more thorough details and description of what was done exactly is needed. It is unclear what inputs and outputs were in all cases, and how RefFinder was used to determine the most optimal reference genes.
Line 149. “HSP70 gene was used to asses the validity of selected reference genes”
Why was HSP70 chosen to validate? Some rationalization would be helpful. In the last paragraph of the discussion, the motivation becomes clearer, that it could be hypothesized that the heat shock protein expression would be induced by stress, for example, at higher temperatures or after pesticide exposure, but this needs to be explicitly stated at this stage of the manuscript in the methods/results section.
Results.
Section 3.1. Reporting of the results is especially unclear in this section.
From Line 156 to 157, it appears that there is a shift from reporting on PCR/cloning to qRT-PCR, but this transition is not explicitly clear. There needs to be clear text introducing that the qRT-PCR results are now being reported from Line 157 onward.
Lines 157-158. “analysis of the dissolution curves of nine candidate reference genes and one heat shock protein gene”
What dissolution curve? There is no mention of generating dissolution curves in section 2.6 of the Materials and Methods. It needs to be described how this was done. Furthermore, in Table 1 and Table 2, 9 genes total are shown, including HSP70. There should not be 9 reference genes and 1 heat shock protein.
Line 159. “Standard Curve”
What standard curve. Again, in Materials and Methods, there is no mention of what experimental design was used, what dilution series, to generate the standard curve.
Line 190. “Therefore, the number of reference genes in qRT-PCR was two in different stage”.
It is not clear what exactly the meaning of this sentence is.
Figures and Tables.
In Figure 2, in the legend, on Line 233, it says “according to RefFinder”….but in the Figure on the Y axis, it says Geomean of ranking values. In the appropriate materials and methods section it needs to be made clear how this was determined. What was the input for this Software?
In Figure 4, in Panel 4b, there are two charts shown. Nothing is mentioned specifically or clearly in the relevant text of the results (lines 248-256) nor in the Figure 4 legend (lines 266-269) why there are two charts shown in this panel.
In Table 3, it is clear that depending on which software is used, the rankings can vary quite a bit. How is this reconciled. If RefFinder is used to reconcile all of the data, it needs to be made clear in the appropriate Materials and Methods section how this was done exactly.
Discussion.
Lines 290-302. In this paragraph, several times results are compared to a study on cotton aphid. When this comparison is made, a citation/reference is needed here, after such statements, to the cotton aphid study.
Line 297-299. “All analysis programs, including….GAPDH, EF-1a were the most stable reference gene in different wing dimorphism and different temperature.”
This claim seems to be inaccurate. According to Table 3, especially for the temperature experiments, GAPDH ranks only 4th (not most stable) for three of the four different softwares examined.
Comments on the Quality of English LanguageThe manuscript is poorly written, both in terms of language, but also in clarity of the message, throughout the document. Specific examples where clarity is needed are given below. And it is advised that after all revisions are made, that the manuscript be thoroughly language edited before re-submission.
Reviewer 2 Report
Comments and Suggestions for Authors
This article evaluates the suitability of 9 genes as qPCR reference genes. GAPDH and EF-1 β were recommended as the most stable reference genes under different issues in sorghum aphids. The scientific value of a paper is not very significant. In addition, there are many grammar and writing errors in the text.
Comments on the Quality of English LanguageSome grammar errors:
L21 and many other lines: Italic Latin scientific names,
L42: The Sorghum Aphid, The first letter of an English name should not be capitalized
L43: forage crops pest should be forage crop pest
L43: harm Sorghum should be “harming sorghum”
L44: what are swarms? bees? It is better to use colony for aphid.
L80: it should be “Insect rearing” or just “Insects”
Fig 2 The gene names should be italicized
Fig 3 what are v2/3, v3/4……? The authors should provide explanations in figure captions
Fig 4 I suggest using segments in Y axis.
Reviewer 3 Report
Comments and Suggestions for Authors
General comments
The manuscript by Guan et al., aims to define a set of reference genes to be used in RTqPCR experiments in the aphid Melanaphis sacchari, an agricultural pest that causes problems to several cultures, including sorghum and sugarcane. This is a very important resource for those working on the biology of M. sacchari. However, the manuscript fails to address several important aspects. In the introduction, the efforts that have been made towards defining good practices in RT-qPCR experiments are not taken into account (for a recent comment, please see (https://doi.org/10.1016/j.mam.2024.101249)). Insufficient information is provided in the Material and Methods section, which compromises the reproducibility of the reported experiments, and again, fair referencing to relevant papers, has not been made. This is of major concern in a manuscript describing a set of reference genes to be employed in RT-qPCR experiments and which will probably be frequently employed/cited by other researchers that study M. sacchari. In the results section, the overall rationale of the experiments is not clear, and the presentation of the results is compromised. Finally, the Discussion session mainly repeats the description of the results already presented in Results without discussing them in relation to the literature and again, without adequate referencing.
Specific comments
1. Introduction. The authors should revise the cited references.
-Lines 53 to 61. Please cite the work of Bustin and collaborators. Either this recent paper (https://doi.org/10.1016/j.mam.2024.101249) or better still, the original MIQE work, which is an important reference for those working with RT-qPCR (https://doi.org/10.1373/clinchem.2008.112797).
-Line 70 Drosophila Melanogaster, change to Drosophila melanogaster.
-Line 73. The authors indicate a set of nine reference genes that have been used in previous studies as reference genes. One of them is the HSP70 gene, which in the present manuscript has not been used as a reference gene. Instead, the authors investigated the pattern of HSP70 expression in different conditions. This is a point that merits revision. Further, this list of nine reference genes is not accompanied by a reference(s) and the reader cannot identify the original source of the information.
2. Material and Methods
-Item 2.1 please review the information on aphid cultivation, by introducing a description of the life cycle. Part of this information is mainly in lines 88-91, could be moved to item 2.1 and further improved.
-Items 2.2.1 and 2.2.2 please clarify if the three samples for each stage/tissue came from independent biological samples. For example, were all first instar nymphs from the same grafting or at least three independent graftings were used to obtain the three samples? Also please clearly explain the storage conditions.
-Also in item 2.2.2 what do the authors mean by “wing dimorphism”? what is “windless aphids”?
-Item 2.3.1 What was the stage collected here? Adults? Please specify.
Item 2.3.2 The phrase “Thirty aphid after treated by imidacloprid remained to survive, while sixty aphids treated by distilled water and stored as described earlier.” It is not clear to what “earlier” refers to. Please clarify.
Item 2.4. Total RNA extraction and cDNA synthesis. Please clearly explain the method employed for RNA extraction. Which instrument was employed to analyze the quality of the extracted RNA? What was the yield of each sample? Indicate the manufacturer of the PrimeScriptR RT reagent kit. Please indicate the amount of RNA that was employed in the reverse transcriptase reaction. Please clarify if DNAse treatment was performed and in which step.
Item 2.5 Please indicate a reference for the aphid transcriptome mentioned in line 121. Please clarify what is “cloned by traditional PCR” (line 124). Lines 125 and 126 seem to be the title of the next table (Table 1).
Item 2.6 Please indicate the manufacturer of the PCR instrument and provide information of the manufacturer of the plates/tubes employed.
Item 2.6.1 Please cite or review the original reverences for DeltaCt, geNorm, Normfinder, BestKeeper. Not all cited references seem to be correct. Lines 141-145 it is not possible to understand what analysis was actually done. Please clarify and rewrite. Also please include a reference for RefFinder.
Item 2.6.2 It is not clear why HSP 70 is being described in this session because initially (Introduction) it is presented to the reader as a possible reference gene.
Results
Item 3.1 It is unclear why two sets of primers were designed (Tables 1 and 2). Also it is not clear if the products of the primers listed in Table 2 were cloned or if these products were subjected to the analysis that is described. If the products of primers listed in Table 1 were cloned, please justify and describe the results. Additionally, how were the primers listed in Table 2 validated?
Item 3.2 The statement “The expression abundance of candidated reference genes is also the main screening condition for reference gene screening.” needs a reference to support it. Please find a validated and widely accepted reference.
Item 3.3.1 , line 186, what is RefFinder? A reference is needed. Maybe accompanied by further explanation as to the reasons it was used.
Item 3.3.3 how was wing dimorphism evaluated? Why was it evaluated?
Legend of Figure 2. Please describe the graph (F),
Legend of figure 3. The legend needs improvement. There might be some information missing in it.
Item 3.3 Why was HSP70 used as a test gene?
Figure 4. Please improve the title of this figure. There is no mention to the statistical test employed to analyze these results. Please clarify and describe it as well in Materials and Methods. Also please inform what is CK that appears in Figure 4C.
Discussion
Overall the discussion needs to be improved. The results are not compared to previous results, neither in aphids, not in other insect groups. The only exception is the data related to the cotton aphid, for which no reference is provided. Also please cite references regarding hsp70 when comparing the results in the aphid with those in the literature.
-Line 307 it is not clear what the authors mean by “adoption abilities”. Why would it be related to differential expression of HSP70?
-Lines 311 to 314. Please revise the functions of HSP70 and consider rewriting the conclusion regarding the changes in temperature experiments.
Comments on the Quality of English LanguageThe English is very difficult to understand, with the exception of the Simple Summary and Conclusions.
